# Implementation of a complex intervention to improve care for patients whose situations are clinically uncertain in hospital settings: A multi-method study using normalisation process theory

Halle Johnson[1]*, Emel Yorganci[1], Catherine J. Evans[1,2], Stephen Barclay[3], Fliss E. M. Murtagh[4], Deokhee Yi[1], Wei Gao[1], Elizabeth L. Sampson[5], Joanne Droney[6], Morag Farquhar[7], Jonathan Koffman[1]

1 Cicely Saunders Institute, Florence Nightingale Faculty of Nursing, Midwifery & Palliative Care, King's College London, London, United Kingdom, 2 Sussex Community NHS Foundation Trust, Brighton General Hospital, Brighton, United Kingdom, 3 Primary Care Unit, Department of Public Health and Primary Care, University of Cambridge, Cambridge, Cambridgeshire, United Kingdom, 4 Wolfson Palliative Care Research Centre, Hull York Medical School, University of Hull, Hull, United Kingdom, 5 Marie Curie Palliative Care Research Department, University College London, London, United Kingdom, 6 The Royal Marsden NHS Foundation Trust, London, United Kingdom, 7 School of Health Sciences, Faculty of Medicine and Health, University of East Anglia, Norwich, United Kingdom

* halle.johnson@kcl.ac.uk

## Abstract

### Purpose

To examine the use of Normalisation Process Theory (NPT) to establish if, and in what ways, the AMBER care bundle can be successfully normalised into acute hospital practice, and to identify necessary modifications to optimise its implementation.

### Method

Multi-method process evaluation embedded within a mixed-method feasibility cluster rando-mised controlled trial in two district general hospitals in England. Data were collected using (i) focus groups with health professionals (HPs), (ii) semi-structured interviews with patients and/or carers, (iii) non-participant observations of multi-disciplinary team meetings and (iv) patient clinical note review. Thematic analysis and descriptive statistics, with interpretation guided by NPT components (coherence; cognitive participation; collective action; reflexive monitoring). Data triangulated across sources.

### Results

Two focus groups (26 HPs), nine non-participant observations, 12 interviews (two patients, 10 relatives), 29 clinical note reviews were conducted. While coherence was evident, with HPs recognising the value of the AMBER care bundle, cognitive participation and collective action presented challenges. Specifically: (1) HPs were unable and unwilling to

**Data Availability Statement:** All relevant data are within the paper and its Supporting Information files.

**Funding:** This paper presents independent research funded by the National Institute for Health Research (NIHR) under Health Technology Assessment Programme (15/10/17). This research was supported by the National Institute for Health Research Collaboration for Leadership in Applied Health Research South London (NIHR CLAHRC South London), now recommissioned as NIHR Applied Research Collaboration South London. CJE is funded by a HEE/NIHR Senior Clinical Lectureship (ICA-SCL-2015-01-001). The views and opinions expressed are those of the authors and do not necessarily reflect those of the NHS, the NIHR, MRC, NETSCC, the NIHR or the Department of Health. The funders had no role in study design, data collection and analysis of the study.

**Competing interests:** The authors have declared that no competing interests exist.

operationalise the concept of 'risk of dying' intervention eligibility criteria (2) integration relied on a 'champion' to drive participation and ensure sustainability; and (3) differing skills and confidence led to variable engagement with difficult conversations with patients and families about, for example, nearness to end of life. Opportunities for reflexive monitoring were not routinely embedded within the intervention. Reflections on the use of the AMBER care bundle from HPs and patients and families, including recommended modifications became evident through this NPT-driven analysis.

## Conclusion

To be successfully normalised, new clinical practices, such as the AMBER care bundle, must be studied within the wider context in which they operate. NPT can be used to the aid identification of practical strategies to assist in normalisation of complex interventions where the focus of care is on clinical uncertainty in acute hospital settings.

## Introduction

Poor hospital care has received growing attention, particularly among the frail elderly and those approaching end of life [1]. There is increasing recognition of the challenges of caring for this growing population, many of whom face clinically uncertain outcomes in which they may improve or deteriorate further [2, 3]. These patients often have complex clinical and psychosocial needs. However, they are often inadequately addressed due to poor identification of deterioration [4] and insufficient and delayed communication from health professionals [4–6]. This has potential to negatively impact on patients and their families at a profoundly emotional level [1]. It also affects health professionals and health systems. It has been argued their intolerance of clinical uncertainty includes sub-optimal decision-making and planning, poor communication, inappropriate levels of investigation, patient safety, and use of scarce healthcare resources [6].

In response, a growing number of interventions, including the Serious Illness Conversation Guide [7, 8], the Universal Form of Treatment Options (UFTO) [9], the Recommended Summary Plan for Emergency Care and Treatment (ReSPECT) [10], and the Psychosocial Assessment and Communication Evaluation (PACE) [11], among others, have emerged. These interventions have been designed specifically to assist health professionals identify patients whose situations are clinically uncertain and more effectively navigate decision-making [10, 12, 13], communication [5, 7, 8] and care for those approaching the end of life. Broadly, these interventions aim to provide a structured approach to managing complex needs and uncertainty by developing a clearly communicated and documented care and treatment plan, that incorporates escalation or de-escalation decisions [14]. They also aim to ensure that patient preferences, in the context of clinical uncertainty, are taken into account, documented and understood across the wider clinical team [3].

However, these interventions are all inherently complex [15]. They are often situated within complicated care settings, require the successful navigation of multiple interacting components, and the involvement of health professionals, across specialties and roles, as well as patients who are acutely unwell and their families [2]. In the context of clinical uncertainty and approaching end of life, this complexity may be exacerbated, with patients, many of whom are acutely unwell, relying on family to make decisions on their behalf [16]. Further,

negotiating critical decisions between health professionals, patients and their families, as well as across clinical specialities, can lead to interactional conflicts, often associated with differing vested interests and professional paradigms [16–18]. If these dynamics are not sufficiently considered and understood, such interventions may be delivered inappropriately with potentially harmful consequences for patients, or not delivered at all [19–21].

The 'AMBER care bundle' (where AMBER refers to Assessment; Management; Best practice; Engagement; Recovery uncertain) is a notable example of these complex interventions. It was developed in 2010 to overcome issues of inadequate and discordant decision-making and communication in the acute hospital setting [3]. It aims to improve care for patients who are deteriorating, clinically unstable with limited reversibility, and at risk of dying in the next one to two months [3]. The latter criterion was subsequently amended to be at risk of dying during a patient's episode of hospital care, despite treatment [22]. The AMBER care bundle is designed to make clinical decision-making explicit in situations of uncertainty by encouraging health professionals to work in concert with patients, who possess sufficient mental capacity, and their families, to develop and document a clear medical plan, including consideration of anticipated outcomes, cardiopulmonary resuscitation and escalation plans, while continuing to acknowledge their situation of uncertainty [12]. The intervention encourages regular communication with the patient and family regarding treatment plans, preferences for care and any other concerns S1 Appendix [12].

A growing body of literature has attempted to shed light on the processes and outcomes associated with the intervention, however, these have demonstrated mixed findings. A comparative observational mixed-methods study of the AMBER care bundle identified increased frequency of discussions about prognosis between health professionals and patients, and higher awareness of prognosis by patients [23]. However, despite instances of communication being greater, they were often associated with a lower quality of information being communicated [10]. Interviews with health professionals in a qualitative study identified that the AMBER care bundle was often utilised as a tool to categorise patients, to change the focus of care delivery and indirectly served a symbolic purpose in influencing behaviours of individuals and teams [24]. More recently, a feasibility cluster randomised controlled trial (cRCT) of the AMBER care bundle across four UK hospital sites revealed a highly varied experience of care and communication for patients supported by the intervention and their relatives [22, 25].

Clinical and contextual equipoise therefore is still present. Concerns are amplified as this intervention has been identified by National Health Service (NHS) England as one of five key enablers to Transform End of Life Care in Acute Hospitals [26], highlighted as representing 'good practice' by both the Royal College of Physicians [27] and the National Institute for Health and Care Excellence [28]. Additionally, it has been widely adopted across a network of approximately 40 UK District General Hospitals [29] and a growing number of hospitals in New South Wales, Australia [30].

## Choice of normalisation process theory for process evaluation

The development of this complex intervention was informed by a pragmatic clinical case note review with input from two specialties–palliative care and geriatric medicine [3]. There is now increasing evidence that the successful development and implementation of complex interventions benefit from the contribution of theoretical frameworks [31–33]. As part of the feasibility cRCT of the AMBER care bundle [22], we conducted a process evaluation to understand how the intervention was operationalised, and what modifications and refinements were needed to optimise its use in acute hospital settings. To achieve this, we made use of Normalisation Process Theory (NPT) [34], a socio-behavioural theory. NPT was chosen as it focuses on the *'social*

*organisation of the work (implementation), of making practices routine elements of everyday life (embedding) and of sustaining embedded practices in their social contexts (integration)'* [34] (p 538).

NPT provides a set of tools to identify and explain the social processes through which new or modified practices of thinking, enacting and organising work are operationalised in institutional settings: in this case, hospitals [21]. Moreover, it sets out a three-stage model of implementation, embedding and integration and is organised around several important questions: (i) *What factors promote or inhibit the routine incorporation of the intervention in practice*? (ii) *What factors promote or inhibit the implementation, embedding and integration of the intervention*? (iii) *What factors promote or inhibit the mobilisation of structural and cognitive resources for the implementation of the intervention*? [35]

The theory identifies four essential determinants of embedding or 'normalising' complex interventions into common practice. These are (i) *'coherence'*–the extent to which an intervention is understood as being meaningful, achievable and valuable; (ii) *'cognitive participation'*–the engagement of individuals (in this case, health professionals) necessary to deliver the intervention; (iii) *'collective action'*–the work that brings the intervention into use; and (iv) *'reflexive monitoring'*–the on-going process of adjusting the intervention to keep it in place [22]. These components are considered to be dynamic and interact within the wider context of the intervention, such as existing organisational structures and procedures [36]. Importantly, they are in keeping with the UK's Medical Research Council (MRC) guidance on the development and evaluation of complex interventions [37] and the Methods of Researching End-of-life Care (MORECare) statement [15], which both stress the importance of theory in understanding what makes interventions effective.

## Aim and objectives

In this paper, we examine the use of NPT to determine if, and in what ways, the AMBER care bundle can be successfully embedded or 'normalised' into acute hospital care practice to support patients whose situations are clinically uncertain, and their families. More specifically, the paper aimed to integrate our data across sources under NPT constructs to (i) understand individual and contextual facilitators and barriers surrounding the implementation of the AMBER care bundle and (ii) identify strategies to strengthen facilitators and mitigate barriers, informing the optimisation of the intervention and its future sustainability in acute hospital clinical care.

## Methods

### Design

We made use of a multi-method design [38] within the wider mixed-method feasibility cRCT trial [22]. Data were collected in parallel and then analysed and integrated using NPT. The data collection approaches included (i) health professional focus groups, (ii) interviews with patients and/or relatives, and (iii) multi-disciplinary team meetings' non-participant observations. The quantitative component involved a detailed examination of patients' clinical notes. The study was registered in a freely accessible clinical trial registry (ISRCTN36040085).

### Study setting

The feasibility cRCT took place across purposefully selected general medical wards located in clusters. For the purposes of this study, the clusters were represented by four district general hospital (DGHs) in England. Implementation of the AMBER care bundle were limited to two of the four DGHs. Selection of study wards at each site was informed by heat maps that

**Table 1. Description of study sites.**

| Site | Cluster | Specialty | Number of beds | End of life care plan | CQC rating |
|------|---------|-----------|----------------|------------------------|------------|
| **Site 1** | 1 general medical ward | • respiratory<br>• endocrinology | 30 | Individualised care plan for dying patients | Good |
| **Site 2** | 2 general medical wards | • care of the elderly | 36 | End of life care plan | Requires improvement |

provided contextual information at a ward level on the number of deaths during and up to 100 days after admission. Wards with the highest number of deaths per year were considered to be suitable for the study [22, 29]. In this paper, we focus on the two intervention sites, where the AMBER care bundle was implemented and delivered and who provided data for the process evaluation reported in this paper Table 1. Full details of how the sites were selected are reported elsewhere [22].

## Implementation of the AMBER care bundle

A nurse facilitator supported the implementation of the AMBER care bundle across the two intervention sites for a period of two months. This involved: familiarisation with the ward, introducing the intervention to health professionals and training them on its use, supporting them in the practice of using the intervention, and observing how they used it in practice. Full details of the implementation of the intervention across study sites is reported in detail elsewhere [22, 29].

## Sampling

**Recruitment of patient and family for interviews.** Patient or family participants were identified by research nurses in conjunction with health professionals for the feasibility cRCT when patients met the following criteria: were over 18 years old, deteriorating, in a clinically uncertain situation with limited reversibility, at risk of dying during their current episode of care, despite treatment. Participants also needed to be able to provide written informed consent or assent through a personal consultee (consultee declaration) prior to the interview. We made the decision to pragmatically stop recruitment when we believed we had collected an adequate amount of data to address the research questions and where we could be confident from our on-going interviews and processes associated with our framework analysis approach that new data would be considered to be redundant of data already collected [39–41]. Each participant provided written informed consent prior to the interview.

**Recruitment for focus groups.** Ward staff from study wards were invited via research nurses and posters to participate in the focus groups. Of those who expressed interest, we aimed to recruit a range of health professionals with different levels of experience. Written informed consent was obtained from participating health professionals.

**Recruitment for non-participant multi-disciplinary observations.** For non-participant observations, the researcher (EY) organised with research nurses and clinical staff an appropriate schedule of attendance. Observations included multi-disciplinary team meetings including morning handovers and board rounds across study wards. Informed consent was obtained prior to these meetings.

## Data collection

Questions asked during (i) focus groups with ward staff, (ii) semi-structured interviews with patients and/or relatives; (iii) issues noted during the non-participant observations of multi-

disciplinary team meetings; and, (iv) the review of patient participants' clinical notes, were informed by our patient and public involvement (PPI) representatives, and were aligned to NPT.

**(i) Qualitative interviews with patient and carers.**   Interviews were semi-structured and topic guides explored key constructs of NPT including patients' and their relative's insights into the delivery of care S2 Appendix. Interviews were conducted by a research assistant (EY) and recorded on an encrypted digital voice recorder. During transcription, all potentially identifiable information was removed or anonymised.

**(ii) Focus groups with health professionals.**   Health professionals' views on caring for patients whose situations are clinically uncertain and views about the intervention were explored during focus groups S3 Appendix. Focus groups were led by senior researchers (JK and CE) with experience in palliative care and qualitative research, and field notes were taken by EY and HJ. Focus groups were audio-recorded and transcribed verbatim and anonymised.

**(iii) Non-participant observations of multi-disciplinary team meetings.**   Non-participant observations of multi-disciplinary team meetings took place at multiple time points on each of the wards. During these observations, the researcher (EY) noted who was present, the frequency and length of the meetings, and the type of conversations relating to patients identified as fulfilling the criteria to be supported by the AMBER care bundle. Notes were also made as to which professionals contributed to conversations, and the decision-making discussion and actions related to patient care. Observations were recorded as hand-written field notes throughout and immediately on leaving the meeting.

**(iv) Patient participant clinical case note review.**   Following the implementation of the AMBER care bundle, we examined the clinical notes of patients who were in receipt of the intervention. Data were extracted by EY onto a designed-for-purpose form which captured details of admission, death or discharge, calculation of the length of stay, and documentation of the intervention components. All identifiable patient information was removed or anonymised before sharing with the rest of the research team.

## Analysis

**Patient participant clinical case note review.**   The numerical data in the case notes were analysed with SPSS [42] using descriptive statistics.

**Interviews with patients and relatives, focus groups with health professionals, and non-participant observations of multi-disciplinary team meetings.**   Qualitative data were analysed using the Framework approach and thematic analysis [41] facilitated by NVIVO 10. Members of the research team who led on conducting the interviews, focus groups and observations (JK, EY and HJ) also led on the analysis. They familiarised themselves with the raw data and discussed their impressions of the dataset. The NPT constructs–'coherence', 'cognitive participation', 'collective action' and 'reflexive monitoring'–provided a thematic framework Table 2. We took a robust approach to analysis: all interviews and focus groups were double coded in NVIVO 10 by two researchers (EY and JK, EY and HJ, or JK and HJ) independently followed by comparing results and discussion within each researcher pair to ensure uniformity of coding. We also hosted 'data workshops' where the researchers coded a sample of transcripts together with our patient and public involvement (PPI) members to minimise bias in interpretation and the validity of findings. Once transcripts were coded, data was exported from NVIVO 10 and were charted and mapped using Excel®. Non-confirmatory data were also explored and consideration was made about their sources to avoid making unwarranted claims about patterns in the data [43].

**Table 2. NPT constructs relevant to the AMBER care bundle.**

| Normalisation Process Theory (NPT) Constructs | NPT framework questions relevant to AMBER care bundle |
|---|---|
| **NPT construct 1- 'Coherence'**<br>The work people engage individually and collectively when they are faced with the problem of operationalising a set of practices | • Is the AMBER care bundle easy to describe?<br>• Is it distinct from other ward-based interventions?<br>• (i.e., meaning and sense-making by participants) Does the AMBER care bundle have a clear purpose for all relevant participants i.e. ward staff?<br>• Do ward staff have a shared sense of its purpose?<br>• What benefits will the AMBER care bundle bring, and to whom?<br>• It is AMBER care bundle expected to improve the performance and the clinical outcomes of patients and their families.<br>• Are these benefits likely to be valued by potential participants?<br>• Does the AMBER care bundle fit with the overall goals and activity of the organisation? |
| **NPT construct 2-'Cognitive participation'**<br>'Buy-in' or relational work people do to build and sustain a community of practice around a complex intervention. | • Do ward staff consider the AMBER care bundle to be a good idea?<br>• Will they see the point of the AMBER care bundle easily?<br>• Will ward staff be prepared to invest time, energy and work in it? |
| **NPT construct 3-'Collective action'**<br>The operational work that people do to enact a set of practices around a complex intervention. | • How will the AMBER care bundle affect the work of ward staff?<br>• Will it promote or impede their work<br>• Will ward staff require extensive training before they can use it? |
| **NPT construct 4-'Reflexive monitoring'**<br>The monitoring work that people do to understand and appraise the ways that a new set of practices affect them and others around them. | • How are ward staff likely to perceive the AMBER care bundle once it has been in use for a while?<br>• Will the AMBER care bundle to be perceived as advantageous for patients or ward staff?<br>• Will it be clear to them what the effects of the AMBER care bundle intervention have been?<br>• Can users/staff contribute feedback about the AMBER care bundle once it is in use?<br>• Can the AMBER care bundle intervention be adapted/improved based on experience? |

Data across all sources were then discussed further by the researchers and triangulated using the NPT constructs to understand the operation of the AMBER care bundle on each ward. At this stage, researchers identified areas of confirmation and contradictions across sources which were used to greater researchers understanding of the operation of the AMBER care bundle across health professionals, and the contributing facilitators and barriers involved in the sustainable use of the intervention.

## Research governance and ethical approval

Ethical approval was obtained from the National Research Ethics Committee—Camden and King's Cross (20.12.2016, REC Reference: 16/LO/2010) and Health Research Authority (25.01.2017). Local research governance approvals were obtained from participating hospitals.

## Results

We conducted two focus groups (26 health professionals), nine non-participant observations, 12 interviews (two patients, 10 relatives) and 29 patient participant clinical note reviews. Demographics of those involved in focus groups, interviews, non-participant observations and clinical note reviews are provided in S1–S4 Tables respectively. Using multi-methods, the implementation process across sites based on the four NPT constructs of coherence, cognitive participation, collective action and reflexive monitoring were examined. Within each construct, we present the barriers and facilitators to implementation and discuss strategies for optimising implementation of this complex intervention within the acute hospital setting.

### Coherence–making sense of, and finding meaning in, the AMBER care bundle

Coherence represented the process through which ward staff shared a common and valid interpretation of the purpose and value of the AMBER care bundle. Overall, ward staff were observed as having a good practical understanding of the intervention and its constituent components. For example, during the non-participant observation of a morning handover at Site 1, a range of health professionals were noted as being confident and clearly explaining the intervention to a new consultant on the ward.

There was broad agreement from health professionals that the intervention represented a positive shift in the emphasis of care for a patient group who were previously overlooked in clinical practice. Ward staff recognised that the intervention prompted them to recognise and prioritise patients whose situations were clinically uncertain and further, engage in important discussions with them regarding preferences for care, escalation decisions and medical treatment. Many noted the value of this for ensuring that patient and family preferences were discussed, captured and communicated with those involved in the patients' care in a timely manner.

The AMBER care bundle was also perceived to be valuable in supporting some staff to provide care for this patient group. At Site 2, the ward manager highlighted the value of the intervention for junior staff. First, in increasing their understanding of clinical uncertainty, deterioration, and end of life and secondly, their increased confidence to engage with this patient population by using the intervention as a platform to broach such topics.

> . . . But now with the 'AMBER' I think they can talk, and they will feel more confident to talk with relatives.
>
> (Site2-014, female, Ward Manager)

Whilst most health professionals suggested that the AMBER care bundle represented a fundamental change in care, a small number did not believe it differed noticeably from their existing practice. These individuals did, however, note that the intervention acted as a means to formalise their current practice, a view typified by the following junior doctor:

> Speaking for myself, if I have someone who I'm worried and has the potential of deteriorating, I would always within my best capacity to try and inform the family about their situation. I don't think it changed our practice. The only thing we're doing is just formally documenting it. We were doing everything we could, uhm yeah, even prior to AMBER bundle.
>
> (Site2-005, Female, Senior House Office)

Despite health professionals holding a coherent view of the value of the AMBER care bundle, a lack of clarity surrounding the intervention's eligibility criteria resulted in a varied understanding of what patients were most appropriate for the AMBER care bundle. In practice, the clinical team were frequently observed making judgements on patients' suitability to be supported by the intervention, based on the patients' level of co-morbidity, frailty, disease progression, the likelihood of responding to treatment and medication, or their 'ceiling of care escalation', that represented proxies for 'clinical uncertainty'. They rarely referred to a patient's 'risk of dying' during the episode of care [the admission] to inform decisions. This was further exemplified by the responses received by health professionals during focus groups who when asked to describe the patients who were suitable to be supported by the intervention used descriptors such as 'those who are aged and frail', or 'those with an unpredictable recovery'.

One junior doctor on Site 2, explained that they had focused on identifying patients in clinically uncertain situations rather than those at risk of dying, due to the latter requiring them to prognosticate, which they did not feel skilled or confident to do. This posed fundamental challenges and often made this aspect of the eligibility criteria impossible to operationalise coherently. Further, this reluctance to acknowledge a patient's 'risk of dying' was noted by patients and their family, who were often more aware of their own, or their loved one's likelihood of death.

*I remember having a conversation with the doctor and saying, "Do you really actually think he's going to be discharged out of here? Because he looks like he's a dying man to me." The doctor just said to me "You have to be optimistic", and I just said, "Optimistic or realistic?" you know?*

*(Site1-017, Carer)*

## Cognitive participation—Commitment and engagement with the AMBER care bundle

Cognitive participation represented the extent to which key stakeholders (health professionals) were adequately motivated to incorporate this complex intervention into their practice and how well it fitted in with existing approaches. All staff were expected to engage in the active identification of patients whose situations were clinically uncertain, discuss plans for treatment and care with patients and their family, and document these in patients' clinical notes.

Across sites, staff participating in the focus groups were unanimous that the role of the nurse facilitator was critical in successfully engaging with and operationalising the intervention. She encouraged them to appraise which patients might be suitable for the AMBER care bundle, as well as, importantly, prepare and reflect on conversations with patients and their families which were often difficult and emotionally demanding. Observations of multi-disciplinary team meetings supported these views. At numerous points, the nurse facilitator was observed encouraging health professionals to complete AMBER care bundle components, particularly in identifying instances of clinical uncertainty, ensuring important discussions took place with patients and families and reminding staff of the requirement to document these circumstances.

*He's 'AMBER', so while writing the discharge letter, we should remember to note the things discussed and escalation decisions.*

*(FIELD NOTE: Site2-011, Female, Nurse Facilitator: observed during Handover meeting)*

Importantly, because of their pressured workloads, ward staff did not believe they had additional capacity to 'champion' the intervention in the ward within their roles. During focus groups, they questioned their ability to engage in the delivery of the intervention without the continued dedicated support from the nurse facilitator. Specifically, they highlighted the challenges and time required to train new staff about the intervention, a situation amplified by increasing levels of staff turnover.

Staff were fearful of potential negative consequences which may come from the use of the intervention without a dedicated 'champion' to facilitate engagement. At the very least, there was concern that they would overlook patients who might benefit from the intervention and, more seriously, a wariness of engaging in potentially emotionally laborious conversations with patients and or their relatives.

*I think the nurse facilitator was excellent actually in helping us implement and 'do the AMBER'. Since she's gone, I think it dropped off a little bit.*

*(Site1-022, Female, Ward Manager)*

## Collective action—Work required to make the AMBER care bundle function

Collective action represented the notion that ward staff performed actions or tasks based on principles outlined by the intervention. The data from across the different components of the study shed light on the resources and procedures associated with its integration into routine practice. Since the general principle of delivering patient-centred care was shared across staff, for some, the intervention was not perceived to represent a radical departure from their current way of delivering care for some individuals. This consequently meant that the work associated with the collective delivery of the intervention was generally well accepted into daily clinical practice.

Nevertheless, formally operationalising the daily clinical activities associated with the intervention provided a welcomed opportunity for other ward staff to be involved in discussions about the decision-making process associated with the delivery of care for this patient group. Instances of teamwork and shared decision making were observed during the handover meetings. At Site 2, for instance, a range of health professionals contributed to discussions about patients' suitability for the AMBER care bundle. In one focus group, a consultant also highlighted that whilst it was doctors who were typically perceived as being pivotal in patients' care, the views of other professions increasingly began to contribute to patient-centred decisions and their medical plans. In this respect, the presence of the intervention enabled and empowered those who had previously not been called on for their views, to actively share them. A consultant on Site 1 emphasised the shift from clinical decisions being made independently to decisions made as an MDT, with health care assistants who often have more insight into the day-to-day care of the patient, and allied health professions now being involved in decision making.

Perspectives from some patients and relatives substantiated health professionals' views, highlighting teamwork across professionals and specialties during their care, or the care of their family member:

*100% work together well.*

*(Site2-009, Male, Patient)*

*The OTs especially have been brilliant there . . . and yeah they seem to, they seem to all know what's not and what's going on.*

*(Site2-016, Female, Carer)*

The case note review of patient participant notes provided further insight into ward staffs' involvement in patient-centred decisions and medical plans Fig 1. In Site 1, 90% of patient participants' notes detailed discussion and agreement of patient medical plans with ward staff, and in Site 2, this was noted in 67%, demonstrating documented involvement of other health professionals.

Further analysis of the patients' clinical notes provided additional insight into the collective action taken to complete the required documentation associated with the delivery of the intervention. Overall documentation of medical plans and escalation plans for patients across sites were high (medical plans 90% and 78% and escalation plans 80% and 78%, for Site 1 and 2 respectively).

This high level of compliance may be due to health professionals perceiving the documentation required as part of the AMBER care bundle as valuable, simple, and easy to complete.

Despite this, there were, however, collective action-related concerns with some of the intervention procedures, particularly those that relied on timely discussions between health professionals and patients and/or their relatives. The combined findings from the focus groups, interviews with patients and relatives, and review of clinical notes identified that the completion of initial conversations and follow-ups with patients or their family were often delayed. In focus groups, health professionals reflected that this was often because many of the patients suitable for the intervention were elderly, confused and lacked mental capacity. Consequently, engaging in conversations with these patients about their situation, and making treatment and care plans, necessitated the presence of family.

Relatives, however, were not always available to have important discussions in a timely manner. For instance, some were unable to visit or were only able to visit on weekends or evenings, when health professionals most familiar with the patient's circumstances were not on the ward. Observations of multi-disciplinary team meetings further evidenced the challenges ward staff reported experiencing in accessing some family members. For instance, when

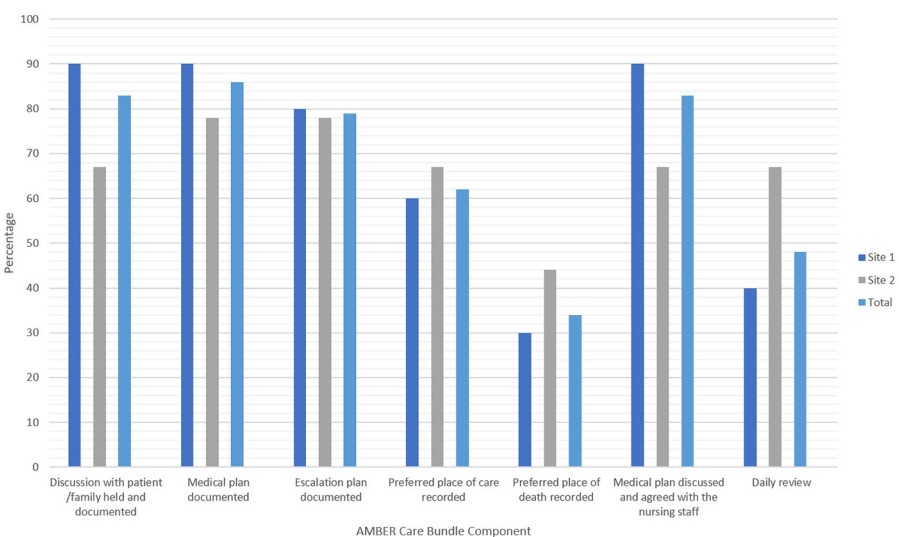

**Fig 1. Documentation of AMBER care bundle components in patient participants' clinical notes.**

talking about a patient on Site 1, a consultant at the multi-disciplinary team meeting was over-heard suggesting a conversation was required with the patient's wife to discuss the possibility of a referral for a dementia assessment. However, an occupational therapist mentioned that the patient's wife was herself unwell and currently receiving chemotherapy, so any discussion soon would be unlikely.

A key component and goal of the AMBER care bundle is the requirement for staff to document patients' clinical situations and medical plans to ensure all staff, including weekend and out of hours staff, are familiar with a patients' situation. While patient participant clinical notes highlighted that this documentation was occurring in most cases, the detail in these notes often varied from patient to patient, translating to instances of inconsistency in care.

This is exemplified by a daughter of a patient on Site 1 who, when asked if she felt like she was getting consistent information from nurses and doctors, highlighted the importance of all staff being aware of a patient's clinical situation to ensure they are receiving consistent care.

*I think probably some of the nurses, I don't want to criticise them but I think probably some of the nurses who were maybe, bank nurses and things, who didn't know her [the patient] so well didn't quite realise about the delirium and thought it was the dementia.*

(*Site 1–002, Carer*)

Similar experiences were shared by other relatives. Another daughter of a patient on Site 1, highlighted instances where her father had been in receipt of unnecessary tests from some health professionals who did not seem aware of the de-escalation for his clinical situation.

During engagements with patients and relatives, health professionals were encouraged, as part of the intervention, to discuss future plans, including preferred place of care and preferred place of death. Despite this, the clinical notes of patients demonstrated that these preferences were often inadequately recorded, particularly place of death. This was documented in only 30% and 44% of patient clinical notes for Site 1 and Site 2, respectively. Compliance with these components of the intervention appeared to be lower than other care plan elements, such as completion of escalation plans and do not attempt resuscitation (DNAR) discussions. This corresponded with the views of patients and relatives who suggested that, while other areas of care were being discussed (such as treatment), preferred place of care and preferred place of death were often absent from these conversations.

*No, we didn't have any of those conversations in the hospital, and to be honest I haven't had any of those conversations since he's gone to the nursing home, which we need to have really to be honest.*

(*Site 1–017, Carer*)

One consultant explained that discussions surrounding the preferred place of care or pre-ferred place of death were difficult for health professionals, due to concerns they might not fulfil patients' or relatives' expectations if preferences could not be realised due to over-riding clinical reasons. One consultant explained that for some patients, while the preferred place of death is home in reality, it is unlikely that these patients would be able to be supported at home, due to the services they need and care they would require. There was therefore reluctance to engage in these conversations with patients in fear that they may be disappointed if preferred place of care or death could not be achieved.

Findings also highlighted that communication with patients and families was often variable due to workload pressures. During conversations, staff felt rushed and unable to provide

adequate time to have these important discussions with patients and their relatives. This proved to be frustrating and health professionals believed that, whilst the intentions of the intervention were honourable, the fidelity of its delivery was often compromised. This was typified in the following comment from a registrar:

> *Sometimes our communication is brief, just because of the time pressures we are in nowadays. Even when I have the conversation, I sometimes feel that if I spent 10 minutes more or 15 minutes more with a patient, I'd probably explain a lot better.*

> *(Site2-002, Male, Registrar)*

**Communication and daily review.** Engaging in difficult conversations was also heavily dependent on health professionals' skills and confidence. This varied considerably. While some consultants reported possessing well-honed skills and expertise in this area, other staff, particularly nurses, believed their skills were inadequate. One ward manager reflected that this was due to nurses having inadequate training in advanced communication, whereas doctors have more focus on this during their training.

Due to this perceived disparity in skills and confidence of ward staff, there were repercussions associated with the delivery of core collective activity in planning care and treatment in concert with patients and relatives. Since some ward staff were reticent to engage in these challenging conversations, this task was delegated by default to consultants, often leading to delays in these discussions occurring, due to consultants already pressured workloads. This finding was supported by the interviews with relatives who were frustrated by the lack of information provided by nurses, health care assistants and other front-line staff, particularly as they were told they would need to wait for the consultant or doctor to provide an update.

> *The nurses never really actually talked about his, erm, terminal diagnosis . . ., it was only the doctors, I'm sure the nurses were aware. . . his erm, condition would change, but the nurses didn't really talk about that.*

> *(Site 1–001, Carer)*

> *Most of them were healthcare assistants but they were able to say functionally able to say how he was, but if you asked them anything medically, they wouldn't answer your questions.*

> *(Site 1–001, Carer)*

The emotive nature of discussions relating to clinical uncertainty and the impact of these on some health professionals, particularly junior doctors', also caused issues.

> *I think as a med student, we didn't have to broach that with the family, so we just meet the patient and put them on 'AMBER care'. So, I'm meeting you for the first time and they may cry, and you know I found it quite hard, emotionally.*

> *(Site 2–023, M, Foundation Doctor Y 1)*

While the nurse facilitator provided some support in this respect, there was a strong desire to provide greater access to clinical supervision and protected time for staff to debrief after these disconcerting discussions.

The daily review component of the intervention also experienced low compliance; this intervention component was documented in only 40% and 67% of clinical notes at Site 1 and Site 2, respectively. This component of the intervention required ward staff to take time each

day to, first, review whether a patient's situation of clinical uncertainty remained and, second, to follow-up on any patient or family concerns and preferences identified during the initial discussion. Two issues were identified by health professionals to explain variance in the daily review. Firstly, some health professionals were of the view that appraising patients' clinically uncertain recovery status daily was unnecessary. This was exemplified by a consultant at Site 1 who preferred for his team to feel that there was always an element of uncertainty and to review patients less often, unless there is a drastic change in their situation of clinical situation. Secondly, others believed that frequent discussions with patients and families about their situation of clinical uncertainty had the potential to cause additional and undue distress.

As part of the daily review, staff were also required to place a yellow sticker on patients' clinical notes each day that they were on the AMBER care bundle, alongside any relevant notes. In practice, the 'A' stickers were not perceived as adding value, were burdensome and therefore were quickly abandoned by staff.

However, in contrast to health professionals' concerns surrounding the daily review, relatives reported valuing the frequent updates from health professionals. Accepting the extensive workload and time pressures of health professionals, some suggested that they did not expect lengthy discussions, but brief updates would have been welcomed each time when they visited the ward.

*We wanted more communication. We were there every day, so there was no reason why they did not stop and spoken to us.*

*(Site1-003,Carer)*

### Reflexive monitoring—Opportunities to appraise the AMBER care bundle

Reflexive monitoring refers to the appraisal process which health professionals, formally and informally, undertook to assess and understand the ways that the AMBER care bundle affected themselves, patients and/or families and others around them. Opportunities for formal reflection and monitoring were not integrated into the intervention. However, we did identify, through observations and focus groups, attempts by health professionals to locally modify or reconstruct the AMBER care bundle and its delivery, to successfully enable implementation within their acute hospital setting. This included adapting the frequency of reviewing patients' situation of clinical uncertainty and removal of the requirement to place a yellow 'A' sticker in patients' notes Table 3.

**Optimising implementation, integration and sustainability of the AMBER care bundle.** Fig 2 presents a model of the facilitators and barriers of the AMBER care bundle alongside modifications that must be considered to enable the normalisation of the intervention within the acute hospital setting. In response to these barriers, improvement strategies are identified that are likely to contribute to improving aspects of cognitive participation and collective action to sustain practices associated with the delivery of this complex intervention [14, 44].

## Discussion

Introducing new models of working in health care settings such as hospitals is often extremely challenging [45]. This multi-method process evaluation study, residing within a wider cluster cRCT, has demonstrated a varied response to implementing the AMBER care bundle into routine practice to serve patients whose situations are clinically uncertain and their families. Whilst it is evident from our findings that the intervention was perceived as enhancing some aspects of care for this previously under-served patient group, we also identified barriers that

**Table 3. Modifications to the AMBER care bundle component.**

| Suggested Modification | Rationale for Modification |
| --- | --- |
| **Remove prognostication from eligibility criteria** | Health professionals highlighted the difficulty of predicting whether patients were going to die during their current hospital admission. Consequently, many were reluctant to make decisions on patients' suitability for the AMBER care bundle based on their risk of death and instead focused on identifying situations of clinical uncertainty to inform their decisions. Additionally, health professionals suggested that simplification of the eligibility criteria to concentrate solely on 'clinical uncertainty' rather than 'deterioration' and 'risk of dying' would not only ensure that a wider group of patients would be identified and benefit from the AMBER care bundle, but it would mean that staff would not be required to use the ambiguity of prognostication as a decision-making tool. |
| **Removal of daily review stickers** | Health professionals saw little value in the requirement of placing a yellow 'A' sticker delineating 'AMBER' on patients' clinical notes to prompt staff to think about their situation. In practice, this task associated with the intervention was rarely completed. Health professionals therefore recommended that the sticker should be disposed with. |
| **Daily review of the patient's situation of clinical uncertainty** | Health professionals suggested that reviewing patients' clinical uncertainty within the clinical team was not required daily since patients' situations did not tend to change between recovery and deterioration that often. Further, some health professionals perceived that the requirement to revisit conversations on a daily basis were distressing for patients and family members. Staff therefore recommended that it would be more valuable and efficient to review patients' clinical situations only where there was evidence of a more profound change in their situation. |
| **Daily re-engagement with patients and/ or family** | Paradoxically, patients and particularly relatives suggested that staff should provide a brief practical update to the patient and family each day regarding their general overall care. Aware of workload pressure of staff, patients and relatives suggested that these did not need to be lengthy discussions. Instead, they should be brief, covering any notable events that occurred throughout the previous day and provide a general update regarding their care. It was suggested that these brief updates could be undertaken by nurses and other ward staff who were present more often on the wards. |

block its successful integration and normalisation as it was intended and consequently its sustainability in routine practice. Below, we critically reflect on the findings to understand the intervention through an NPT lens, describing which aspects of the intervention work, which did not and importantly, why. We also outline what is vital to improve the intervention so it can be normalised and sustained in 'real-world' acute hospital ward settings and deliver favourable patient and family outcomes.

## Aspects that aided normalisation of the AMBER care bundle

The intervention was used by health professionals to prompt them to actively consider the care and management of a patient group which were not previously prioritised in their clinical practice. The intervention itself acted as a platform to broach the issue of clinical uncertainty within the acute hospital setting, helping to shift the focus of care provided. This symbolic use of the AMBER care bundle is in line with previous qualitative explorations which identified health professionals use of interventions to prompt changing the focus of care [24, 46]. For some health professionals, the AMBER care bundle was perceived to differ little from their current ways of working. Considering the growing number of end of life care-related support aids or tools, for example ReSPECT [10], a lack of differentiation may be present and problematic, with

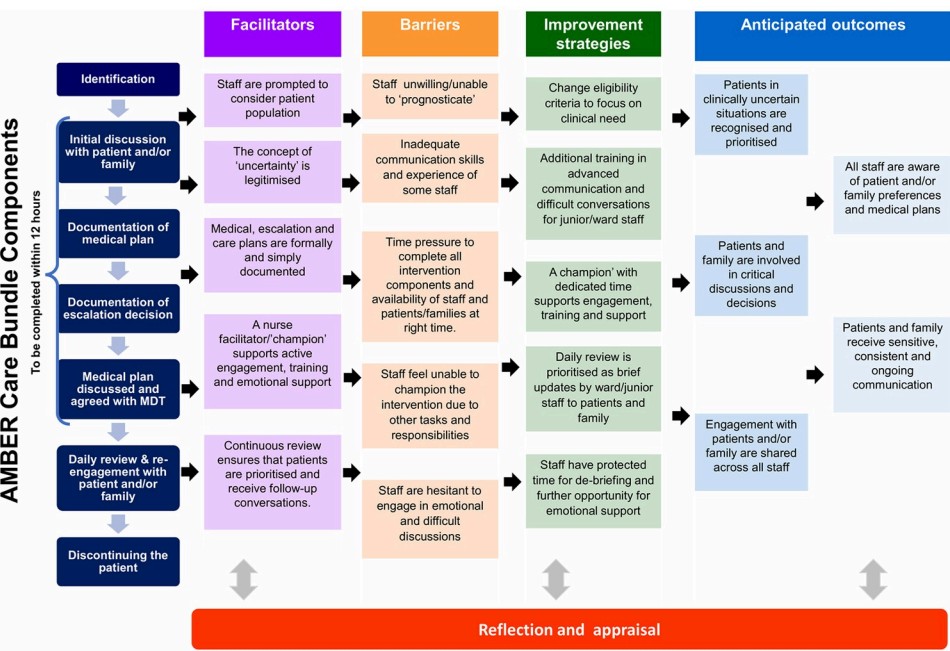

**Fig 2. Facilitators, barriers and strategies for normalisation of the AMBER care bundle in acute hospital settings.**

some staff resisting interventions they believe may replicate existing practices and reflect an unnecessary administrative burden [14]. Yet most health professionals perceived little administrative burden associated with the AMBER care bundle, with the required documentation considered simple, clear and easy to complete. Further, while health professionals recognised the value of the AMBER care bundle for improving their own processes (simple and clear documentation, the involvement of all staff in decision making), they were also clear on the value and benefit of the intervention for patients and their families, including empowering patients to share their preferences earlier, and involving patients and their family in critical decisions. This is a welcomed shift away from an overemphasis on processes to a greater focus on the ultimate goals of intervention delivery: improved care for patients and their families [46].

Critical to initial and sustained engagement with the intervention was the provision of a 'champion' (nurse facilitator) who ensured that health professionals were prompted to consider all components of the intervention and were supported throughout delivery. The importance of a 'champion' has been previously emphasised for the successful implementation of integrated end of life pathways within organisations [10, 47]. For the AMBER care bundle, the 'champion' promoted active engagement, trained new staff members and provided emotional support to health professionals delivering the intervention.

## Aspects that were disadvantageous to the normalisation of the AMBER care bundle

Staff shared views on their concerns associated with identifying patients suitable for the AMBER care bundle which currently require staff to judge a patient's 'risk of dying during their episode of care' alongside their clinical uncertainty and deterioration. In practice, health professionals made decisions about who was suitable for the intervention based more broadly on their capacity to benefit from it, rather than specifically their 'risk of dying'. This highlights health professionals unwillingness to make judgements based solely on prognostication and is

in keeping with current evidence which notes the difficulty some health professionals have recognising impending death [48]. Further, research suggests these judgements are highly subjective and are frequently inaccurate, which raises important issues for the operationalisation of the intervention [48]. Solutions to this issue were present during the development of ReSPECT [10]. During the development phase of the intervention designers incorporated cognitive interviewing to understand exactly how terms were interpreted across different professional groups and levels of seniority in the field, prior to its wider-scale implementation [10]. The results of this work contribute to ensuring intervention fidelity and also offer insights into where a more nuanced, flexible interpretation of the intervention eligibility criteria may be required. In its current form, we provide evidence that the eligibility criteria of the AMBER care bundle warrants serious consideration since it limits opportunities to potentially escalate care for those patients who might benefit from critical or high dependency care settings [49], as was originally intended by this intervention [3]. This has potential implications for patient experience and outcomes.

Inadequate skills and confidence in communication also led to issues in the delivery of the intervention. Variation in staff skills meant that the delivery of difficult and daily follow up conversations with patients and their family were frequently allocated to specific health professionals to complete (those with greater experience and confidence in communicating), creating workload pressures and delays in delivery. Further, the emotive context surrounding difficult conversations relating to clinical uncertainty has the potential to impact on health professionals' well-being, particularly among some junior doctors [50]. This is pertinent in light of the ongoing global COVID-19 pandemic [51] where a growing number of relatively inexperienced junior doctors are being called upon to engage with profoundly unwell patients whose situations may be clinically uncertain, in addition to their very distressed families. Caring for patients in this situation is especially challenging, requiring additional skills, excellent communication, and importantly, systems of support for those delivering care [16, 52]. The absence of such systems may contribute to health professionals experiencing what has been referred to as 'moral injury' [53], representing deviations or transgressions from health professionals' moral beliefs and expectations that are witnessed, perpetrated, or allowed by that individual. When not acknowledged and supported, this can lead to stress, depression, anxiety and post-traumatic distress.

Contextual issues, particularly workload pressures and time constraints were also cited as a significant barrier to the delivery of intervention components. Similar to previous studies exploring the implementation of complex interventions into clinical practice [47, 54, 55] frequent staff-turnover and competing priorities led to challenges and meant that health professionals felt unable to 'champion' the intervention themselves, raising concerns about the sustainability of the intervention.

Notably, many of the patients who the health professionals believed suitable to be supported by the AMBER care bundle often lacked mental capacity. This was associated with additional time needed to contact the family to be involved in decision-making, creating further workload pressures, and often resulting in delays in important discussions. Having a clear understanding of the population the intervention intends to serve is critical to minimise harmful outcomes and is emphasised in the Medical Research Council and MORECare guidance [15, 37, 56].

## Supporting normalisation of the AMBER care bundle in acute hospital settings

To support the normalisation of the AMBER care bundle in acute hospital wards, important considerations regarding the skills, knowledge and attitudes of those delivering the intervention,

and an understanding of the context in which it exists, is required. First, staff need to have a coherent view of who is suitable for the AMBER care bundle. It is therefore imperative that the AMBER care bundle eligibility criteria focus solely on a patient's 'clinical uncertainty' and their clinical needs. Criteria that avoid the need for prognostication may ensure patients who would benefit from being supported by the intervention can be quickly, and more efficiently, identified by health professionals [16].

Second, staff need to be equipped with adequate communication skills to engage clearly and compassionately with patients and their family and are allocated protected time to debrief following difficult encounters. Advanced communication training and integrated opportunities for emotional support therefore need to accompany the AMBER care bundle to ensure staff are skilled to deliver the intervention effectively. This is imperative for nursing and ward staff who are often more present on wards and therefore have potentially more opportunities to communicate with patients and their families. Additionally, previous research highlights the need to ensure this training is continually re-visited to further develop and strengthen staff knowledge and skills [50, 57, 58].

Last, while patients' clinical uncertainty may not change on a day-to-day basis, and the clinical team should adapt the frequency of reviewing this to their local needs, it is imperative that brief daily updates for patients and their family are still prioritised. These brief, but frequent updates should aim to provide patients and their family with an update on any medical and/or care changes and provide an opportunity for patients and family to discuss any changes in preferences.

## Implications for clinical practice

The AMBER care bundle carries the potential to address a crucial gap in clinical practice and further refinements, as suggested above, would help ensure it is adequately normalised into practice. Patients and their families, who experience uncertainty in light of their conditions, appreciated communication from health professionals and found the opportunity to discuss issues that matter to them valuable. One advantage of the AMBER care bundle is the fact that it relies on the shared human experience of uncertainty. For health professionals, this uncertainty is about the prognosis and likelihood of imminent death or the possible outcomes of the active treatment. Clinical uncertainty can exist for many conditions and has been argued as being an inevitability [59]. While emotive discussions, linked to expressing clinical uncertainty, have generally been seen as the job of designated health professionals, the recent COVID-19 global pandemic [60], has brought with it instances where more health professionals are required to do this work. Specifically, the COVID-19 pandemic has witnessed health professionals dealing with an extraordinary number of cases in acute hospital settings where clinical uncertainty is omnipresent across multiple levels including within the health-system, among professionals, and the patients and their families, for which the course of disease deterioration and potential for recovery is still relatively unknown [51, 61, 62]. As such, with considered modification, the AMBER care bundle has potential to offer health professionals with an approach to better serve patients affected by the COVID-19 and their families.

## Strengths and limitations

Our approach used multiple data sources to inform our findings. This represented a practical and feasible way to explore NPT constructs associated with the implementation and operationalisation of the AMBER care bundle in acute hospital care settings, with qualitative components permitting us to explore salient contexts and mechanisms in more detail. Furthermore, the use of four data sources, and data being analysed iteratively by an inter-professional team,

including input from our PPI members, should increase the dependability of the study findings and interpretations.

The use of the four NPT constructs as an analytic framework enabled us to provide an understanding of how the AMBER care bundle did, and in many instances could not become normalised within an acute hospital setting. Associated with this, we also identified barriers and facilitators to the future successful integration and implementation of the AMBER care bundle.

This study is also subject to limitations. It represents one component of a wider feasibility cRCT of the AMBER care bundle and was restricted to just two study sites.

Second, we are also mindful of the absence of nursing representation in one of the focus groups, who may have provided additional insights into the use of the intervention in practice from this professional group. Although, several nurses expressed interest, and confirmed their availability beforehand, on the day of the focus group no nurses were able to attend, due to urgent clinical commitments. This highlights the issues faced while researching in a 'real-world' context and could be overcome in future studies by considering additional flexibility and resources in the study design to accommodate the unpredictable nature of clinical work. For example, holding two smaller health professional focus groups at each site, or following up individually with health professionals who were unable to join focus groups at the allocated time.

Third, we were not able to conduct direct observations of care due to 'real-world' resource and logistical constraints associated with the feasibility study. Direct observations of clinical practice would be possible if research ethical approval for the study included a provision for consent at the level of the unit/cluster level [63].

Last, whilst the model has been informed by NPT constructs, developed as a result of detailed discussions within the research team, the modifications suggested as a result of this study to the AMBER care bundle have yet to be tested in the field.

Future exploration of the AMBER care bundle across other care settings and professional groups will be valuable in providing further understanding of the normalisation of this intervention in practice.

## Conclusions

Our findings support growing evidence that to be successfully implemented, scalable and to be of value, new clinical practices such as the AMBER care bundle must consider the social, organisational and environmental context in which they are required to operate [45]. Whilst individual health professional change is necessary, the local context in which the intervention is intended to operate must also be supportive when implementing new, often highly complex, clinical interventions. Omitting this has potentially direct implications for patient and family experience and outcomes, patient safety and staff well-being, including issues raised by the *Independent Review of the Liverpool Care Pathway* [64] that specifically stressed the importance of understanding interventions focused on clinical uncertainty and communication when caring for the dying.

The importance of in-depth examination of implementation processes should proceed with feasibility studies of complex interventions to identify and incorporate modifications required so that the intervention operates as intended. Our findings highlight both facilitators and barriers and offer practical strategies for normalising multiple inter-related components of a complex intervention where the focus of care is on clinical uncertainty and end of life care in acute hospital settings. This has particular resonance during a time when the global COVID-19

pandemic is challenging patient care, shared decision-making and planning, and exercising health professionals in an unprecedented manner [62].

It is central for the normalisation and successful sustainability of such interventions that the health professionals who deliver the intervention feel empowered, and supported, in contributing reflexively to making recommendations about the workability of intervention. As the AMBER care bundle is already being operated in some DGHs, key stakeholders involved in implementation development must be receptive to these findings and scrutiny. The costs of not doing so are now regrettably well known in palliative and end of life care.

## Supporting information

**S1 Appendix. The AMBER care bundle tool.**
(DOCX)

**S2 Appendix. Topic guide for semi-structured interviews with patient and carers.**
(DOCX)

**S3 Appendix. Topic guide for focus groups with health professionals.**
(DOC)

**S1 Table. Demographics of health professionals involved in focus groups at each site.**
(DOCX)

**S2 Table. Demographics of patients and carers involved in qualitative interviews at each site.**
(DOCX)

**S3 Table. Demographics of health professionals involved in non-participant observations at each site.**
(DOCX)

**S4 Table. Demographics of patient participants involved in clinical case note review.**
(DOCX)

## Acknowledgments

The authors wish to acknowledge support from the research managers at the NIHR Evaluation Trials and Studies Coordinating Centre. We are very grateful to the AMBER Care Bundle design (Dr Irene Carey, Dr Adrian Hopper Susanna Shouls and Charlene Davis) team who offered valuable advice during the study. We wish to thank Linda Launchbury, the nurse facilitator, who represented the nurse facilitator at the intervention sites. We wish like thank the clinical staff at each of the four hospital sites for agreeing to be involved in this study and for participating in the focus groups. We would especially like to thank the principal investigators and the clinical teams at each the four hospital sites: Dr Henry Penn–Northwick Park Hospital, London North West University Healthcare NHS Trust, Dr David Brooks–Chesterfield Royal Hospital, Chesterfield Royal Hospital NHS Foundation Trust, Dr Natalie Broomhead–East Surrey Hospital, Surrey and Sussex Healthcare NHS Trust, and Dr Carole Robinson–Tunbridge Wells Hospital, Maidstone and Tunbridge Wells NHS Trust for agreeing to be involved in this trial. We express particular thanks to the research study nurses working across each of the four study sites who were instrumental in recruiting patients, collecting data, and arranging interview appointments. We are grateful to Sylvia Baily (SB) and Collen Ewart (CE) who provided expert PPI input at all stages of the study.

## Author Contributions

**Conceptualization:** Catherine J. Evans, Stephen Barclay, Fliss E. M. Murtagh, Deokhee Yi, Wei Gao, Jonathan Koffman.

**Data curation:** Halle Johnson, Emel Yorganci, Catherine J. Evans, Jonathan Koffman.

**Formal analysis:** Halle Johnson, Emel Yorganci, Catherine J. Evans, Stephen Barclay, Jonathan Koffman.

**Funding acquisition:** Catherine J. Evans, Stephen Barclay, Fliss E. M. Murtagh, Deokhee Yi, Wei Gao, Jonathan Koffman.

**Investigation:** Halle Johnson, Emel Yorganci, Catherine J. Evans, Jonathan Koffman.

**Methodology:** Catherine J. Evans, Stephen Barclay, Fliss E. M. Murtagh, Deokhee Yi, Wei Gao, Jonathan Koffman.

**Project administration:** Halle Johnson, Emel Yorganci, Elizabeth L. Sampson, Joanne Droney, Morag Farquhar, Jonathan Koffman.

**Validation:** Halle Johnson, Emel Yorganci, Catherine J. Evans, Jonathan Koffman.

**Writing – original draft:** Halle Johnson, Jonathan Koffman.

**Writing – review & editing:** Halle Johnson, Emel Yorganci, Catherine J. Evans, Stephen Barclay, Fliss E. M. Murtagh, Deokhee Yi, Wei Gao, Elizabeth L. Sampson, Joanne Droney, Morag Farquhar, Jonathan Koffman.

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
