## [Decision Letter · Decision Letter 0]

31 Jul 2020

PONE-D-20-11538

Implementation of a complex intervention to improve care for patients whose situations are clinically uncertain in hospital settings: a mixed method study using normalisation process theory

PLOS ONE

Dear Dr. Johnson,

Thank you for submitting your manuscript to PLOS ONE. After careful consideration, we feel that it has merit but does not fully meet PLOS ONE’s publication criteria as it currently stands. Therefore, we invite you to submit a revised version of the manuscript that addresses the points raised during the review process.

We look forward to receiving your revised manuscript.

Kind regards,

Alessandra Solari, M.D.

Academic Editor

PLOS ONE

Journal Requirements:

2. Please ensure that you refer to Figure 2 in your text as, if accepted, production will need this reference to link the reader to the figure.

3. Please include a copy of Table 4 which you refer to in your text on page 20.

Reviewers' comments:

Reviewer's Responses to Questions

**Comments to the Author**

1. Is the manuscript technically sound, and do the data support the conclusions?

Reviewer #1: Yes

Reviewer #2: Yes

2. Has the statistical analysis been performed appropriately and rigorously? 

Reviewer #1: N/A

Reviewer #2: N/A

3. Have the authors made all data underlying the findings in their manuscript fully available?

Reviewer #1: Yes

Reviewer #2: No

4. Is the manuscript presented in an intelligible fashion and written in standard English?

Reviewer #1: Yes

Reviewer #2: Yes

5. Review Comments to the Author

Reviewer #1: There are no statistical concerns as there is very limited to none quantitative analyses. However, the authors may want to clarify the quantitative phase of the mixed methods approach. The quantitative data collection and analyses is unclear and study seems more of a qualitative study.

Reviewer #2: Thank you for asking me to review this very interesting and well-written paper. I really enjoyed reading it.

For the editor’s benefit I should make clear my conflicts of interest at the outset. My first conflict of interest is that I have led the programme of empirical research and theoretical development that led to Normalisation Process Theory, which is employed as the conceptual framework in this manuscript. My second conflict of interest is that I have also jointly led a programme of research in the closely related area of treatment escalation plans that has used a similar method (notes review doi: 10.1136/bmjopen-2018-022021; and qualitative process evaluation https://www.sciencedirect.com/science/article/pii/S0277953620303622) formed around a similar theoretical framework.

My comments on this paper are as follows,

1. Introduction

1.1. This is a very interesting and well written paper which deals with a highly complex problem: forming and shaping decision-making processes as people with complex diseases that sometimes include problems of decisional capacity progress towards end of life.

1.2. The AMBER Care bundle is a widely used to support professionals, patients and families in this context. I think it would be helpful to the reader to do more work at the front end of the paper that describes both the intervention, and the ensemble of practices that using it puts in train. The introduction assumes a fairly well-informed reader. I think you can also frame this as a representative of a class of interventions (decision-making tools) that have a complex and contested history in end of life care – AMBER (along with UFTO and ReSPECT) make sense only if we know a little of the of the interactional context missing here.

1.3. I wonder if it might also be helpful to consider very briefly in the introduction some of the interactional problems that follow from negotiating decisions about end of life care. There is a large literature on this, and I think it would be helpful to think about the context in which AMBER and is operationalised. These a both organisationally and interactionally fragmented, and they are definitely not cognitively or affectively neutral. Again, this is context is important if the reader is to make sense of the assertion that clinical and contextual equipoise is present.

2. Methods

2.1. This is a really interesting study design, that could do with just a little elaboration. My concern here is that it is stated that this is a mixed methods study in which qualitative and quantitative data were collected in parallel and then a form of integrative analysis was undertaken. It is not clear to the read what the quantitative data is. Was it extracted from the notes review. Or, is the mixed methods design the feasibility trial plus the qualitative data. I’m not sure which it is. This is a simple matter to solve.

2.2. Qualitative data was collected from two sites. Although it’s stated that details of site selection are available elsewhere, it would be useful to have at least a precis of this here. This is important because the qualitative work reported here is drawn from two sites only. Are these the only sites at which qualitative data were collected? Or are some data (e.g. notes review) drawn from all sites? Was there a comparative element to site selection (e.g. good and bad CQC ratings)?

2.3. My understanding is that ‘saturation’ is a criterion applied to interpretive data collection in grounded theory studies, especially those using the constant comparative method. But you also say that you used framework analysis. I think we just need a couple of sentences to say how using these to techniques together worked.

3. Results

3.1. In the introduction to results you again say that a mixed methods study was performed. Again, I can see that several qualitative techniques were used, but conventionally ‘mixed methods’ is used to signify qualitative plus quantitative data and analyses so this is something that needs to be made clear.

3.2. I like the results section very much. I think the really important result that you draw attention to is that a new category of patient—the ‘clinically uncertain’ patient—is defined and brought into service, as a proxy for the patient who is likely to die. We found something similar in our study of the ReSPECT Treatment Escalation Plan, which is that when our key informants talked about clinically uncertain patients their examples were of people they believed were likely to die. Although I think that prognostic uncertainty is important here, and alternative interpretation is that these tools are intended to be used by different professional groups and hierarchies and that the category ‘clinically uncertain’ permits a high level of flexibility about their use. So, implementing the decision-tool depends to some extent on implementing a new category of patients. (This speaks to the NPT sub-construct of differentiation.)

4. Discussion

4.1. I wonder if Table 5 should be in the discussion rather than the results section?

4.2. I really like figure 2 and I really like table 3: I did wonder if they overlapped a bit , and whether it would work better to just have figure 2?

4.3. In the strengths and limitations section it is stated (i) that restricting the study to two sites, and (ii) the absence of nursing representation in a focus group limit the transferability of the findings. I’m not sure that they do. This study explores a common problem and its management. Janice Morse introduced the concept of ‘theoretical generalisability’ in qualitative research, and well theorised studies of common problems often have a very high level of transferability. After all, we’re still using fundamental concepts from Glaser and Strauss 60 years after their empirical research on awareness contexts and the dying patients was conducted in single wards at three general and one VA hospital in San Francisco.

6. PLOS authors have the option to publish the peer review history of their article (what does this mean?). If published, this will include your full peer review and any attached files.

Reviewer #1: No

Reviewer #2: No

---

## [Author Response · Author response to Decision Letter 0]

13 Aug 2020

Response to Review Questions: 

1. Have the authors made all data underlying the findings in their manuscript fully available?

The PLOS Data policy requires authors to make all data underlying the findings described in their manuscript fully available without restriction, with rare exception (please refer to the Data Availability Statement in the manuscript PDF file). The data should be provided as part of the manuscript or its supporting information or deposited to a public repository. For example, in addition to summary statistics, the data points behind means, medians and variance measures should be available. If there are restrictions on publicly sharing data—e.g. participant privacy or use of data from a third party—those must be specified.

Reviewer #1: Yes

Reviewer #2: No

In response to reviewer 2, we note that we have included sufficient extracts both in tables and within the manuscript to replicate and support our study findings, as such we consider our manuscript to be compliant with the PLOS data policy. 

Response to Review Comments: 

Reviewer 1: 

1. There are no statistical concerns as there is very limited to none quantitative analyses. However, the authors may want to clarify the quantitative phase of the mixed methods approach. The quantitative data collection and analyses is unclear and study seems more of a qualitative study.

Thank you for offering your time to provide a detailed review of our paper and for raising this important issue. Our process evaluation took place within a mixed-methods feasibility cluster randomised controlled trial, and used multi-methods including the following qualitative components (i) focus groups; (ii) semi-structured interviews; and (iii) non-participant observations of multi-disciplinary team meetings. The quantitative component involved a detailed examination of patient’s clinical case notes, to provide an understanding of compliance with each component of the AMBER care bundle. On reflection, we believe using the term ‘multi-methods’ is more appropriate for our study and made this amendment to the title, abstract, and methods section (pg. 7) of the revised manuscript and have subsequently referred to this throughout the manuscript.

Reviewer 2: 

1. Thank you for asking me to review this very interesting and well-written paper. I really enjoyed reading it.

For the editor’s benefit I should make clear my conflicts of interest at the outset. My first conflict of interest is that I have led the programme of empirical research and theoretical development that led to Normalisation Process Theory, which is employed as the conceptual framework in this manuscript. My second conflict of interest is that I have also jointly led a programme of research in the closely related area of treatment escalation plans that has used a similar method (notes review doi: 10.1136/bmjopen-2018-022021; and qualitative process evaluation https://www.sciencedirect.com/science/article/pii/S0277953620303622) formed around a similar theoretical framework.

Thank you for your comprehensive and helpful review of our paper, and for sharing some of your work which we have found most helpful to reflect on and have now referred to this where appropriate when revising our manuscript (please refer to pages 3- 4 in the introduction, and pages 27-28 in the discussion of the manuscript). 

2. This is a very interesting and well written paper which deals with a highly complex problem: forming and shaping decision-making processes as people with complex diseases that sometimes include problems of decisional capacity progress towards end of life.

Thank you very much for your favourable view of our paper. 

3. The AMBER Care bundle is a widely used to support professionals, patients and families in this context. I think it would be helpful to the reader to do more work at the front end of the paper that describes both the intervention, and the ensemble of practices that using it puts in train. The introduction assumes a fairly well-informed reader. I think you can also frame this as a representative of a class of interventions (decision-making tools) that have a complex and contested history in end of life care – AMBER (along with UFTO and ReSPECT) make sense only if we know a little of the of the interactional context missing here.

Thank you for your comment, on pages 3-4 of the revised manuscript we have now provided an additional paragraph in our introduction highlighting the challenges of decision making and communication for those in uncertain situations, and towards the end of life. We have referenced a number of examples of the increasing number of tools responding to these challenges and have highlighted how the AMBER care bundle fits among them. 

4. I wonder if it might also be helpful to consider very briefly in the introduction some of the interactional problems that follow from negotiating decisions about end of life care. There is a large literature on this, and I think it would be helpful to think about the context in which AMBER and is operationalised. These a both organisationally and interactionally fragmented, and they are definitely not cognitively or affectively neutral. Again, this is context is important if the reader is to make sense of the assertion that clinical and contextual equipoise is present.

Thank you for this suggestion, we agree this would be a helpful addition to our introduction. We have now included additional contextual information at the start of our introduction located on page 3-4 of the revised manuscript, referencing both the Campling et al., 2018 and Higginson et al, 2016 papers among others, who highlight some of the difficulties of negotiating decisions at end of life between and across health professionals, patients and their families. 

5. This is a really interesting study design, that could do with just a little elaboration. My concern here is that it is stated that this is a mixed methods study in which qualitative and quantitative data were collected in parallel and then a form of integrative analysis was undertaken. It is not clear to the read what the quantitative data is. Was it extracted from the notes review. Or, is the mixed methods design the feasibility trial plus the qualitative data. I’m not sure which it is. This is a simple matter to solve. 

Thank you for raising this issue. On page 7 of the revised manuscript, we have now clarified in our methods that this process evaluation sat within a mixed methods feasibility cluster randomised controlled trial. Specifically, data collection for the process evaluation involved multi-methods, including qualitative components (the qualitative interviews, focus groups, non-participant observations of multi-disciplinary team meetings and a quantitative component (the examination of the patients’ clinical notes). 

6. Qualitative data was collected from two sites. Although it’s stated that details of site selection are available elsewhere, it would be useful to have at least a precis of this here. This is important because the qualitative work reported here is drawn from two sites only. Are these the only sites at which qualitative data were collected? Or are some data (e.g. notes review) drawn from all sites? Was there a comparative element to site selection (e.g. good and bad CQC ratings)?

Thank you for your suggestion, we have now added further detail on the ‘heat maps’ used to inform study ward selection on page 7 in the methods section of revised manuscript. 

7. My understanding is that ‘saturation’ is a criterion applied to interpretive data collection in grounded theory studies, especially those using the constant comparative method. But you also say that you used framework analysis. I think we just need a couple of sentences to say how using these to techniques together worked.

Thank you for raising this issue. On pages 8-9 of the revised manuscript we now state that we made the decision to pragmatically stop recruitment when we believed we had collected an adequate amount of data to address the research questions and when we could be confident from our on-going interviews and processes associated with our framework analysis approach that new data would be considered to be redundant of data already collected.

8. In the introduction to results you again say that a mixed methods study was performed. Again, I can see that several qualitative techniques were used, but conventionally ‘mixed methods’ is used to signify qualitative plus quantitative data and analyses so this is something that needs to be made clear.

Thank you for raising this, on reflection we agree that this should be referred to as multi-methods and we have adapted this throughout the manuscript. 

9. I like the results section very much. I think the really important result that you draw attention to is that a new category of patient—the ‘clinically uncertain’ patient—is defined and brought into service, as a proxy for the patient who is likely to die. We found something similar in our study of the ReSPECT Treatment Escalation Plan, which is that when our key informants talked about clinically uncertain patients their examples were of people they believed were likely to die. Although I think that prognostic uncertainty is important here, an alternative interpretation is that these tools are intended to be used by different professional groups and hierarchies and that the category ‘clinically uncertain’ permits a high level of flexibility about their use. So, implementing the decision-tool depends to some extent on implementing a new category of patients. (This speaks to the NPT sub-construct of differentiation.)

Thank you for raising this important point. We acknowledge that the concept of clinical uncertainty is nuanced and potentially open to wide interpretation. On page 28 of the revised manuscript we refer to the Hawkes et al. 2020 study – ‘Development of the Recommended Summary Plan for Emergency Care and Treatment (ReSPECT)’ highlighting the importance of intervention designers to conduct cognitive interviews to understand how key terms and concepts are understood so that modifications or refinements can be made to ensure intervention components can be operationalized with fidelity, or to understand where greater flexibility is needed. 

10. I wonder if Table 5 should be in the discussion rather than the results section?

Thank you for making this suggestion. Whilst we understand why it may be reasonable to present this table (previously table 5, now table 3) in the discussion section of the paper, we are minded that the contents of the table reflect suggested modifications resulting from the reflexive monitoring NPT construct. Moreover, in this table we provide contextual information from the perspectives of the health professionals, patients and their relatives as to why these changes were suggested which present key results. We would therefore be very grateful if the Editor and reviewer concede to retaining this table in the results section. 

11. I really like figure 2 and I really like table 3: I did wonder if they overlapped a bit, and whether it would work better to just have figure 2?

Thank you for your comment. We agree that some of the content of the figure and the table are overlapping. The table presents the facilitators, barriers and implications in relation to the constructs of NPT, whereas the figure presents these against the components of the AMBER care bundle. We agree that the figure presents the more valuable contribution to the paper, and by and large contents from table 3 can be ascertained within the main text of the results section of the manuscript. Consequently, we have now removed Table 3. 

12. In the strengths and limitations section it is stated (i) that restricting the study to two sites, and (ii) the absence of nursing representation in a focus group limit the transferability of the findings. I’m not sure that they do. This study explores a common problem and its management. Janice Morse introduced the concept of ‘theoretical generalisability’ in qualitative research, and well theorised studies of common problems often have a very high level of transferability. After all, we’re still using fundamental concepts from Glaser and Strauss 60 years after their empirical research on awareness contexts and the dying patients was conducted in single wards at three general and one VA hospital in San Francisco.

Thank you for your comment and reminding us of this formative study. We agree that the number of sites involved in the study, and absence of nursing representation from one of the focus groups is not a critical limitation of our study. Rather we believe it would have been useful to have more sites and nursing representation to inform potential modifications and improvements to the intervention. On page 32 of the revised manuscript we have changed the text in this section to highlight this. Moreover, we note that future research exploring additional contexts and professional groups may provide additional and potentially useful insights. 

References highlighted in response: 

Campling N, Cummings A, Myall M, Lund S, May CR, Pearce NW, et al. Escalation-related decision making in acute deterioration: a retrospective case note review. BMJ Open. 2018;8(8):e022021.

Higginson IJ, Rumble C, Shipman C, Koffman J, Sleeman KE, Morgan M, et al. The value of uncertainty in critical illness? An ethnographic study of patterns and conflicts in care and decision-making trajectories. BMC Anesthesiol. 2016;16(1):11.

Hawkes CA, Fritz Z, Deas G, Ahmedzai SH, Richardson A, Pitcher D, et al. Development of the Recommended Summary Plan for eEmergency Care and Treatment (ReSPECT). Resuscitation. 2020;148:98-107.

---

## [Editor Report · Decision Letter 1]

2 Sep 2020

Implementation of a complex intervention to improve care for patients whose situations are clinically uncertain in hospital settings: a multi-method study using normalisation process theory

PONE-D-20-11538R1

Dear Dr. Johnson,

We’re pleased to inform you that your manuscript has been judged scientifically suitable for publication and will be formally accepted for publication once it meets all outstanding technical requirements.

Kind regards,

Alessandra Solari, M.D.

Academic Editor

PLOS ONE

---

## [Editor Report · Acceptance letter]

7 Sep 2020

PONE-D-20-11538R1 

Implementation of a complex intervention to improve care for patients whose situations are clinically uncertain in hospital settings: a multi-method study using normalisation process theory 

Dear Dr. Johnson:

I'm pleased to inform you that your manuscript has been deemed suitable for publication in PLOS ONE. Congratulations! Your manuscript is now with our production department. 

Kind regards, 

on behalf of

Dr. Alessandra Solari 

Academic Editor

PLOS ONE